# Identification of a Novel HBV Encoded miRNA Using Next Generation Sequencing

**DOI:** 10.3390/v14061223

**Published:** 2022-06-05

**Authors:** Vladimir Loukachov, Karel A. van Dort, Louis Jansen, Henk W. Reesink, Neeltje A. Kootstra

**Affiliations:** 1Experimental Immunology, Amsterdam UMC Location University of Amsterdam, Meibergdreef 9, 1105 AZ Amsterdam, The Netherlands; v.loukachov@amsterdamumc.nl (V.L.); k.a.vandort@amsterdamumc.nl (K.A.v.D.); 2Amsterdam Institute for Infection and Immunity, Infectious Diseases, 1105 AZ Amsterdam, The Netherlands; 3Department of Gastroenterology and Hepatology, Amsterdam UMC Location University of Amsterdam, 1105 AZ Amsterdam, The Netherlands; jansen.louis@gmail.com; 4Department of Gastroenterology and Hepatology, Leiden University Medical Center, 2333 ZA Leiden, The Netherlands; hwreesink@gmail.com

**Keywords:** miRNA, hepatitis B virus, chronic hepatitis B, next generation sequencing, liver tissue

## Abstract

Hepatitis B Virus (HBV) encoded miRNAs were previously described and suggested to play a role in HBV replication and pathogenesis. In this study we aim to identify novel HBV encoded miRNAs in plasma and liver tissue samples from chronic hepatitis B (CHB) patients and determine their role in CHB pathogenesis and HBV replication. RNA next generation sequencing was performed on plasma and liver tissue samples from ten CHB patients and uninfected controls. The interaction of the potential miRNA-like structures with the RNA-induced silencing complex (RISC) was determined using RNA immunoprecipitation. Expression levels of the HBV encoded miRNAs were measured in liver tissue samples derived from a conformation cohort. The effect of HBV encoded miRNAs overexpression on HBV replication, expression of predicted target genes, and induction of interferon stimulated genes in cell lines were assessed. Three potential miRNA-like structures transcribed by HBV were identified in liver tissue, of which one miRNA, HBV-miR-6, was recognized using RISC. HBV-miR-6 expression was demonstrated in liver tissue samples from 52 of the 87 CHB patients. HBV-miR-6 levels correlated with hepatic HBV-DNA and plasma HBsAg levels. Overexpression of HBV-miR-6 in vitro did not affect HBV replication, and predicted both target genes expression and interferon stimulated genes expression after stimulation. A potential novel HBV encoded miRNA was identified and validated in liver tissue from CHB patients. It is suggested that HBV-miR-6 may play a role in the process of viral excretion or particle formation in vivo.

## 1. Introduction

Approximately one third of the global population is infected with Hepatitis B Virus (HBV), resulting in approximately 290 million people who are suffering from chronic hepatitis B (CHB), with a yearly mortality rate of 887,000 due to liver cirrhosis and hepatocellular carcinoma (HCC) [1,2].

HBV is a small enveloped DNA virus that infects hepatocytes [3,4] and utilizes the host cell machinery for its replication. In the nucleus of the hepatocyte, HBV persists in a replication intermediate form called the covalently closed circular DNA (cccDNA), which serves as a template for viral production [5]. The HBV cccDNA encompasses four partially overlapping HBV open reading frames (ORF) [5] encoding the viral polymerase, which is essential for reverse transcription of the HBV pregenomic (pg) RNA into partially double stranded DNA [6], the structural proteins, hepatitis B surface antigen (HBsAg) and core antigen (HBcAg), and the accessory protein HBx [7,8], which has a regulatory function and is essential for HBV replication in vivo [9]. The hepatitis B e antigen (HBeAg) is also encoded by the C-ORF, although this protein is not necessary for viral replication and is believed to serve as a modulator of the immune response [10]. 

MiRNAs are small, non-coding RNA molecules which can regulate gene expression at a post-transcriptional level. During miRNA biogenesis the pre-miRNA stem-loop structure is cleaved by the endoribonuclease Dicer resulting in double-stranded RNA between 19 and 25 nucleotides in length. This duplex is subsequently loaded into the RNA-induced silencing complex (RISC), where one of the two RNA strands is degraded. The remaining strand, the mature miRNA, binds to the 3′-UTR or 5′-UTR of target mRNAs which can result in either degradation, deadenylation, repression, or upregulation of protein production. It is estimated that human miRNAs can regulate the expression of approximately 60% of genes [11]. Viruses can also encode for miRNAs to regulate viral replication by altering host gene expression [12,13].

Recently, small noncoding RNAs transcribed from the HBV genome were identified in hepatocellular carcinoma (HCC) tissue, and one of these small noncoding RNAs was identified as an HBV encoded miRNA named HBV-miR-3 [14]. Functional analysis showed that HBV-miR-3 suppressed HBV replication by targeting HBcAg mRNA and pgRNA [14]. Furthermore, it was shown that HBV-miR-3 activated the innate immune response to inhibit HBV replication [15] or suppressed the expression levels of protein phosphatase 1A, which is involved in tumor suppression [16]. 

In the current study, we analyzed plasma and liver tissue samples obtained from HCC free CHB patients for HBV encoded miRNAs. The relationship between the identified novel HBV encoded miRNA and biomarkers of HBV replication was examined. Moreover, in vitro experiments were performed to determine the effect of the HBV encoded miRNA on HBV replication, potential target gene expression, and the innate immune response. 

## 2. Materials and Methods

### 2.1. Study Population 

This study was performed in pretreatment samples of CHB patients who participated in a prospective randomized controlled intervention study, which was previously described in detail [17]. This cohort consisted of 151 HBeAg-negative CHB patients with HBV-DNA levels below 20,000 IU/mL, of which 97 matching plasma and liver biopsy samples were available for analysis. Control plasma samples were obtained from 10 healthy volunteers. Control liver tissue from 13 HBV negative patients undergoing surgical liver resection was obtained from the resected liver tissue of the non-affected, tumor free margin surrounding the pathology [18]. Upon further use all liver samples were stored in RNAlater stabilizing solution (Thermo Fisher Scientific, Waltham, MA, USA). Baseline characteristics of the cohorts are shown in Table 1.

### 2.2. Study Design

The current study comprised two steps: the identification step and the validation step. In the identification step, potential HBV encoded miRNA were identified in matching plasma and liver tissue samples of an identification cohort consisting of 10 CHB patients and 10 control samples using miRNA/small RNA NGS. In total, 5 male and 5 female CHB patients were randomly selected, of whom 5 patients were infected with HBV genotype A virus and 5 patients with HBV genotype D virus. In the validation step, candidate HBV encoded miRNAs were validated in the remaining 87 plasma and liver samples of the CHB cohort.

### 2.3. Small RNA Next Generation Sequencing (NGS)

The small RNA NGS was performed by Qiagen miRNA NGS services using optimized protocols for miRNA-specific isolation which were conducted by Exiqon Services (Qiagen, Copenhagen, Denmark). Total RNA was isolated from 500 μL plasma with proprietary RNA isolation protocol optimized for serum/plasma (no carrier added) (Exiqon Services). Total RNA was eluted in ultra-low volume. A total of 5 μL total RNA was converted into microRNA NGS libraries using the QIAseq miRNA Library Kit (Qiagen, Venlo, Netherlands). For liver samples, 100 ng of total RNA was converted into microRNA NGS libraries using the QIAseq miRNA Library Kit (Qiagen, Venlo, The Netherlands). Adapters containing UMIs were ligated to the RNA and the RNA was converted to cDNA. The cDNA was amplified using PCR (22 cycles for plasma samples and 14 for liver samples) and during the PCR indices were added. The PCR samples were purified, and library preparation QC was performed using either Bioanalyzer 2100 (Agilent, Santa Clara, CA, USA) or TapeStation 4200 (Agilent, Santa Clara, CA, USA). Based on the quality of the inserts and the concentration measurements, the libraries were pooled in equimolar ratios. The library pools were quantified using the qPCR ExiSEQ LNA™ Quant kit (Exiqon, Copenhagen, Denmark). The library pools were sequenced using the Illumina NextSeq500 (Illumina, San Diego, CA, USA) sequencing instrument according to the manufacturer’s instructions. Raw data were de-multiplexed and a FASTQ file for each sample was generated using the bcl2fastq v2.20 software (Illumina, San Diego, CA, USA). FASTQ data were checked using the FastQC tool (http://www.bioinformatics.babraham.ac.uk/projects/fastqc/, accessed on 19 February 2020). Cutadapt v1.11 (Mercator Research Center Ruhr, Bochum, Germany) was used to extract the information of adapter and UMI in raw reads, and output from Cutadapt v1.11 (Mercator Research Center Ruhr, Bochum, Germany) was used to remove adapter sequences and to collapse reads using UMI with in-house script. NGS data are available at the NCBI’s Gene Expression Omnibus database (www.ncbi.nlm.nih.gov/geo/, GSE162149). After size selection, small RNA reads that did not match any human sequences were mapped to the corresponding reference HBV subtype A (Genbank ID X02763 and AF297621) and D genomes (Genbank ID X02496) using CLC Workbench (Qiagen, Venlo, The Netherlands). For mapping the reads to the reference genomes two mismatches were allowed. Hotspots were identified that consisted of sequences with several counts. The hairpin-like RNA secondary structure was assessed using the mfold web server online tool. Potential gene targets of HBV-miR-6 were predicted using the miRDB prediction software database [19,20].

### 2.4. RNA Isolation and RT qPCR

Liver tissue (up to 5 mg) was disrupted and homogenized for 4 × 1 min at 30 Hz using the TissueLyser II (Qiagen, Venlo, The Netherlands) and two 5 mm stainless steel beads (Qiagen, Venlo, The Netherlands). Total RNA from liver samples was isolated using the miRNeasy Micro Kit (Qiagen, Venlo, The Netherlands) and from HepG2 and HepG2.2.15 cells using TriPure™ Isolation Reagent (Sigma-Aldrich, Saint Louis, MO, USA), according to the manufacturer’s protocol. The RNA concentration was measured using the Nanodrop 1000 (Isogen Life Sciences, De Meern, The Netherlands). The cDNA was either synthesized using a miRNA specific stem-loop primer containing a universal reverse primer sequence, or an oligo dT primer using the M-MLV Reverse Transcriptase kit (Promega, Madison, WI, USA), according to the manufacturer’s protocol (Appendix A). RT-qPCR was performed using qPCR and RT-qPCR Systems Kits (Promega, Leiden, The Netherlands) and were run on the LightCycler^®^ 480 Instrument (Roche diagnostics, Almere, The Netherlands) using either specific miRNA forward primers and a universal reverse, or target genes and ISGs specific forward and reverse primers (Appendix A). The qPCR was performed using the following program on the LightCycler: pre-incubation steps, 10 min at 95 °C; amplification steps, 50 cycles of 10 s at 95 °C, 20 s at 58 °C, 30 s at 72 °C; a melting curve was obtained to confirm the purity of the PCR product and the specific primer. 

### 2.5. Cell Culture 

HepG2 and HepG2.2.15 cells were cultured in William’s E medium (Lonza, Basel, Switzerland) supplemented with 10% (*v*/*v*) heat-inactivated Fetal Calf Serum (FCS), 2 mM L-glutamine, 5 mM dexamethasone, penicillin (100 U/mL), and streptomycin (100 µg/mL) at 37 °C and 5% CO_2_. HEK293T cells were cultured in Dulbecco’s Modified Eagle Medium without HEPES (DMEM) (Lonza, Basel, Switserland) supplemented with 10% (*v*/*v*) inactivated FCS, penicillin (100 U/mL), and streptomycin (100 mg/mL), and maintained in a humidified 10% CO_2_ incubator at 37 °C.

### 2.6. RNA Immunoprecipitation Chip (RIP) Assay 

The RIP assay was performed using the EZ-Magna RIP™ RNA-Binding Protein Immunoprecipitation Kit (Merck KGaA, Darmstadt, Germany), according to the manufacturer’s protocol. HepG2 and HepG2.2.15 cells were lysed and miRNAs bound to RISC were immunoprecipitated using an argonaute-2 (AGO2) monoclonal antibody (H00027161-M01, Abnova, Taipei City, Taipei, Taiwan). Immunoprecipitations using either small nuclear ribonucleoprotein U1 subunit 70 (SNRNP70) or a mouse IgG antibody were performed as positive and negative controls, respectively. After immunoprecipitation, RNA was isolated using TriPure (Roche Diagnostics, Almere, The Netherlands) and expression of the miRNAs and U1 snRNA was measured using RT-qPCR. 

### 2.7. HBV-miR-6 Overexpression in Cell Lines

HBV-miR-6 sequences flanked by 100 base pair up- and down-stream, to ensure the correct secondary structure folding and processing using RISC, were cloned into a lentiviral vector in which expression is driven by the Cytomegalovirus promotor using the In-Fusion^®^ HD Cloning Kit (Takara Bio, Kusatsu, Shiga, Japan). Infectious lentiviral vector particles were produced via co-transfection of the lentiviral vector construct containing the HBV-miR-6 sequences or a GFP control (22.6 µg per construct) together with pCMV-VSV-G (8 µg), pMDLgp (14.6 µg), pRSV-Rev (5.6 µg) in HEK293T cells using calcium phosphate as previously described [21]. In brief, plasmid DNA was diluted in 0.042 M HEPES containing 0.15 M CaCl_2_, subsequently mixed with an equal volume of 2× HEPES buffered saline (pH 7.2) incubated at room temperature for 15 min and added to the culture medium. After 24 h incubation in a humidified 3% CO_2_ incubator at 37 °C, the culture medium was replaced and cultures were continued at 10% CO_2_ at 37 °C. The virus was harvested at 48 h and 72 h after transfection and passed through a 0.22 μm filter. 

HBV-miR-6 overexpression in HepG2 or HepG2.2.15 cells was established using lentiviral transduction. In brief, cells were seeded in a 24-well plate. After overnight culture, cells were transduced with a lentiviral vector overexpressing either HBV-miR-6 or GFP. Forty-eight hours after transduction the medium was replaced. Culture supernatant and cells were harvested at day 6 after transduction for RNA isolation. In addition, total RNA was isolated 72 h after transduction to determine the expression levels of HBV-miR-6 predicted target genes using qPCR. 

### 2.8. Induction of Interferon Stimulated Genes

HepG2 cells were transduced using a lentiviral vector overexpressing HBV-miR-6 or GFP. After 48 h, the medium was replaced and the cells were stimulated with either IFNα (500 U/mL), R848 (10 µg/mL), ODN2216 (5 µM), or left unstimulated. Total RNA was isolated 16 h after stimulation and the expression of specific interferon-stimulated genes (ISGs) was measured using qPCR. 

### 2.9. HBV-DNA and pgRNA qPCR 

To isolate HBV-DNA from the supernatant of HepG2.2.15 cells, 20 μL of supernatant was incubated with 5 μL DNase 0.05 U/μL (Promega) for 30 min at 37 °C; subsequently, the DNase was heat-inactivated for 10 min at 65 °C. Thereafter, the supernatant was treated with 5 μL proteinase K 1 μg/mL (Sigma-Aldrich) for 30 min at 37 °C; thereafter, proteinase K was heat-inactivated for 10 min at 95 °C. The quantification of HBV DNA was performed using the PG3-forward and BC1-reverse primers (PG3-forward: 5′-CAAGCCTCCAAG CTGTGCCTTG-3′, nt 1865-1886, BC1-reverse: 3′-GGAAAGAAGTCAGAAGGCAAAAACG-5′, nt 1974–1950). For the measurement of intracellular pgRNA, total RNA was first reverse transcribed using the BC1 primer and subsequently the quantification was performed using qPCR using PG3-forward and BC1-long-reverse primers. The qPCR was performed using the following program on the LightCycler: pre-incubation steps, 10 min at 95 °C; amplification steps, 50 cycles of 10 s at 95 °C, 20 s at 58 °C, 30 s at 72 °C; a melting curve was obtained to confirm the purity of the PCR product and the specificity of the primer. 

### 2.10. Statistical Analyses

The expression levels of either the HBV encoded miRNAs, target genes, and ISG were normalized using either the GAPDH or B-actin housekeeping gene; the relative expression was calculated using the 2^−ΔCt method (= 2^− (Ct miRNA/gene − Ct GAPDH/B-actin)). Groups were compared using a Mann–Whitney U test. Association between miRNA expression levels and available baseline biochemical (alanine aminotransferase (ALT)), virological (HBsAg, plasma and liver HBV-DNA), or liver fibrosis (Ishak score, modified HAI grading) levels was analyzed using a Spearman correlation test. Data were analyzed using Graph Pad software (version 8) and SPSS (version 25). 

## 3. Results

### 3.1. Identification of HBV Encoded Small RNAs in Liver Samples of CHB Patients

Small RNA NGS was performed using total RNA isolated from plasma and liver biopsy samples of the identification cohort. On average 25 million raw reads were obtained for each plasma sample, of which 4.3 million reads remained after correction for length and quality. Approximately 761,100 reads were not mapped to the human reference genome and were subsequently mapped to the corresponding HBV reference genome. For liver biopsy samples, on average 21 million raw reads were obtained, of which 12.7 million reads remained after correction for length and quality. Approximately 330,200 reads did not map to the human reference genome and were subsequently mapped to the corresponding HBV reference genome. In plasma samples, no mapping of reads to hotspots in the HBV genome was observed. In liver samples, reads mapped to several hotspots in HBV were obtained; after assessing the RNA secondary structure, we identified three hairpin-like pre-miRNA structures (Figure 1), which were named HBV-miR-6, HBV-miR-7 and HBV-miR-8. HBV-miR-6 mapped between nucleotides 255 and 325, HBV-miR-7 between 805 and 880, and HBV-miR-8 between 1206 and 1272. 

### 3.2. Interaction of Potential HBV Encoded miRNAs with RISC

To determine whether these HBV encoded pre-miRNA hairpin-like structures were recognized using RISC, a RIP assay was performed using an antibody that binds AGO2, a subunit of RISC that recognizes miRNAs (Figure 2). We observed that all potential HBV encoded miRNAs can be detected in the HepG2.2.15 cells (input control). However, only HBV-miR-6 was detected after AGO2 RIP in HepG2.2.15 cells, but not HepG2. An antibody detecting SNRNP70 was used as a positive control, and U1 snRNA was indeed detected after RIP in both cell lines.

### 3.3. Conformation of HBV-miR-6 Expression in Liver Tissue of CHB Patients

To confirm that HBV-miR-6 is expressed in CHB patients, the expression levels of this miRNA were measured using qPCR in the liver tissue samples of the validation cohort consisting of the remaining 87 CHB patients and 13 HBV negative controls. In 52 CHB patients expression of HBV-miR-6 was detected, whereas the miRNA was not detected in liver tissue samples of the HBV negative controls (Figure 3). Furthermore, HBV-miR-6 was not detectable in the plasma of these patients, indicating levels below the detection limit of the assay. Subsequently, in these 52 CHB patients we investigated the correlation between expression levels of HBV-miR-6 and various baseline biomarkers associated with liver disease or HBV replication using a Spearman correlation test (Table 2). Expression levels of HBV-miR-6 were found to correlate with HBsAg and hepatic HBV-DNA levels, but not with plasma HBV DNA levels, ALT, modified HAI grading, or the Ishak fibrosis score. 

### 3.4. The Effect of HBV-miR-6 on Target Gene Expression Levels and Viral Replication In Vitro

A total of 25 different genes that are potentially targeted by HBV-miR-6 were identified using the miRDB prediction software database [19,20] (Appendix A). The effect of HBV-miR-6 on the expression levels of all these potential target genes was investigated in HepG2 cells transduced with a lentiviral vector expressing HBV-miR-6 or GFP. We observed that mRNA expression levels of none of the target genes differed between HepG2 cells that did or did not overexpress HBV-miR-6 (Figure 4A). 

We also determined whether overexpression of HBV-miR-6 had an effect on HBV replication using direct targeting HBV derived mRNAs. To this extent, HepG2.2.15 cells were transduced with a lentiviral vector expressing HBV-miR-6 or GFP. Overexpression of HBV-miR-6 had no significant effect on HBV replication as determined by intracellular pgRNA levels and the production of encapsidated HBV-DNA in the supernatant (Figure 4B,C). 

HBV-miR-6 might also have an effect on in vivo HBV replication through modulation of innate immune responses as was previously described for HBV-miR-3 [15]. The effect of HBV-miR-6 expression on ISG expression was analyzed in HepG2 upon treatment with either a TLR 7/8 agonist R848, TLR 9 agonist ODN2216, or stimulation with IFNα. ISG expression levels were determined 16 h after stimulation. We observed that overexpression of HBV-miR-6 did not affect the induction of ISG expression levels after stimulation with IFNα, R848, or ODN2216 as compared to the GFP control (Figure 5). 

## 4. Discussion

The current study aimed to identify potential HBV encoded miRNAs using NGS in plasma and liver biopsy tissue derived from CHB patients without HCC. Three novel HBV encoded pre-miRNA-like hairpin sequences were identified in liver tissue of the identification cohort, which were named HBV-miR-6, HBV-miR-7 and HBV-miR-8. Although all of the potential novel HBV encoded miRNAs were detected in HBV expressing HepG2.2.15 cells, only HBV-miR-6 was recognized using RISC. This indicates that HBV-miR-6 may serve as an HBV encoded miRNA, while HBV-miR-7 and HBV-miR-8 were most likely degradation products derived from HBV encoded mRNA. 

In contrast, no HBV encoded pre-miRNA hairpin-like sequences were observed in the plasma samples of the same patients, which may indicate that the HBV encoded miRNAs are not released by hepatocytes, which is why they are not detected in plasma or blood. Remarkably, the previously identified HBV-miR-3 [14] was not detected using NGS in either plasma or liver tissue. HBV-miR-3 was identified in HCC tissue derived from CHB patients, whereas in our study only liver tissue of CHB patients without HCC was used. Therefore, HBV-miR-3 is possibly only expressed by HBV at a late stage of disease when HCC is present, thus explaining why HBV-miR-3 was not observed in our study. 

Hepatic expression of HBV-miR-6 was observed in approximately 60% of the CHB patients from our cohort. We observed that hepatic expression of HBV-miR-6 in these patients correlated with hepatic HBV-DNA levels, which implies that HBV-miR-6 may indeed be an HBV encoded miRNA. Unlike previous observation with HBV-miR-3 [14,15], we were unable to show an effect of HBV-miR-6 on HBV replication and the innate immune response in vitro after 6 h and 16 h. It is possible that this innate response is either faster or takes longer to develop, which explains why no effect was observed between the HBV-miR-A overexpression and the control. This is unlikely however, as previous studies showed that hepatocyte or hepatoblastoma cell lines mount an innate immune response well within 24 h after stimulation [22,23,24,25].

HBV-miR-6 was not detectable in the plasma of CHB patients, indicating levels below the detection limit of the assay. HBV-miR-6 is only expressed in HBV infected liver cells; the function of this miRNA is most likely in the infected cell either to support HBV replication or evade immune detection, which can explain why this miRNA is not abundantly detected in the plasma of CHB patients. This also indicates that HBV-miR-6 will not be suitable as a biomarker for viral activity.

The in vivo correlation between HBV-miR-6 expression levels and HBsAg levels, but not plasma HBV-DNA levels, indeed suggests that HBV-miR-6 is not directly involved in viral replication. However, this may imply that HBV-miR-6 is involved in the regulation of the formation and secretion of subviral HBsAg particles without an effect on production of infectious HBV. HBV is known to excessively produce empty subviral particles, which mainly consist of HBsAg, next to infectious particles. The overproduction of subviral particles over-activates the immune system which eventually leads to T-cell exhaustion; moreover, subviral particles can also act as decoys for (neutralizing) antibodies [26]. We were unable to identify a cellular target for HBV-miR-6 in our in vitro analysis; therefore, we cannot confirm this hypothesis in vitro. Additional research, studying cellular requirements for subviral particle production to determine the role of HBV-miR-6 in this process, is ongoing. 

Virally encoded miRNAs might be good candidates as novel biomarkers for viral replication, especially in novel therapeutics which directly target viral products that are also used as biomarkers (e.g., plasma HBsAg, or HBV-DNA). Moreover, miRNAs are not targeted by these novel therapeutics and are easily measured in plasma. However, HBV miR-6 was not detected in plasma using NGS. 

HBV encoded miRNAs can also be ideal therapeutic targets as these miRNAs are HBV specific and can be used to directly regulate HBV transcripts, and thus viral production. In addition, these HBV encoded miRNAs may be involved in regulating cellular proteins to create a favorable environment for HBV replication or evasions of immune signaling or immune detection. Here we observed that HBV-miR-6 expression levels in the liver were associated with hepatic HBV-DNA which indicates that increased HBV-miR-6 levels are related to higher HBV replication in the liver. Additionally, an association with plasma HBsAg levels, but not plasma HBV-DNA levels, was observed, which suggests that HBV-miR-6 plays a role during excretion or particle formation leading to excess production of HBsAg particles. However, we were unable to identify the mechanism of HBV-miR-6 action in our in vitro experiments and further in depth studies are essential.

## 5. Conclusions

Using small RNA NGS we identified three potential hairpin-like structures, which were named HBV-miR-6, HBV-miR-7, and HBV-miR-8, in liver biopsy tissue. Of these potential HBV encoded miRNA, only HBV miR-6 was able to interact with RISC. The expression levels of HBV-miR-6 can be validated in liver tissue derived from a CHB conformation cohort. These expression levels correlated with hepatic HBV-DNA and plasma HBsAg levels, suggesting a potential role in viral excretion or particle formation.

## Figures and Tables

**Figure 1 viruses-14-01223-f001:**
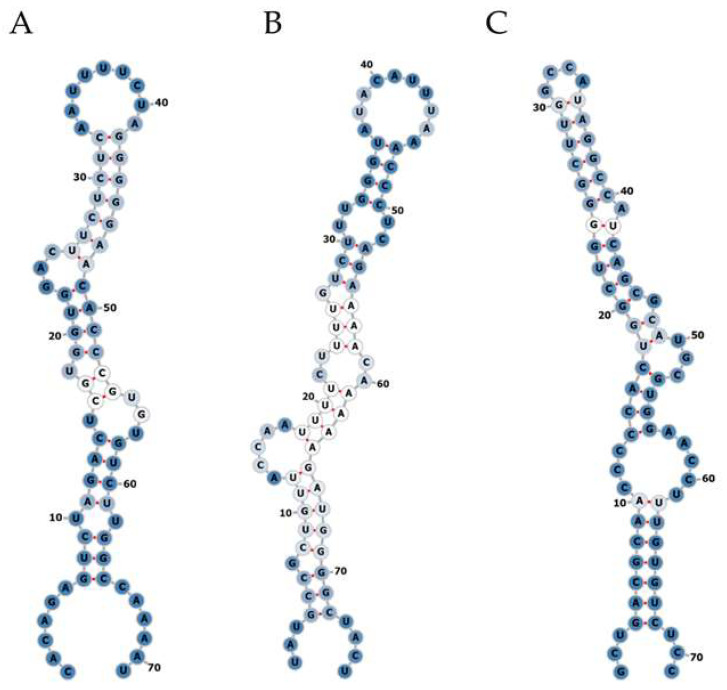
Secondary structure prediction of potential HBV encoded miRNAs. Small NGS of liver tissue from CHB patients revealed three pre-miRNA-like hairpin structures encoded by the HBV genome. The RNA secondary structures of the small RNAs named HBV-miR-6 (**A**), HBV-miR-7 (**B**), and HBV-miR-8 (**C**) were assessed using the mfold web server online tool. A: adenine; C: cytosine; G; guanine; U: uracil. Optimal base pairs are represented by red line between the two nucleotides. The intensity of the blue color represents the level of promiscuity in the association of any given nucleotide or helix with alternative complementary pairs (dark blue been lower and white higher). The numbers indicate the corresponding nucleotide position in the total sequence.

**Figure 2 viruses-14-01223-f002:**
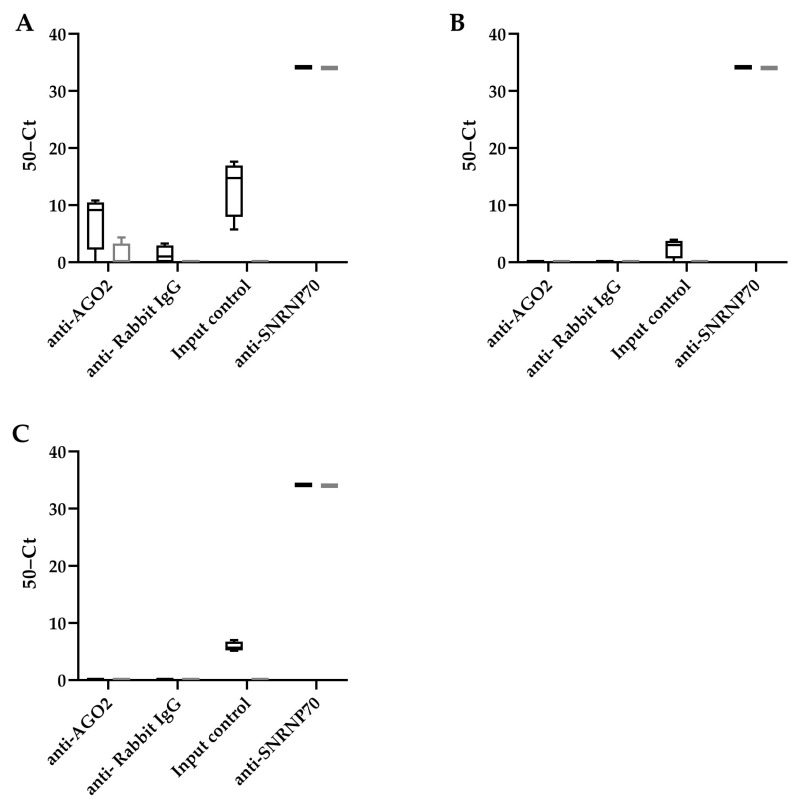
HBV-miR-6 interacts with RICS. RNA immunoprecipitation using either an AGO2, NRNP70 (positive control), or IgG control antibodies was performed in HepG2.2.15 cells (black bars) and HepG2 cells (grey bars). The binding of HBV-miR-6 (**A**), HBV-miR-7 (**B**), and HBV-miR-8 (**C**) to AGO2, and U1 snRNA to SNRNP70, were measured using RT-qPCR. The graphs represent averages of results from two independent experiments.

**Figure 3 viruses-14-01223-f003:**
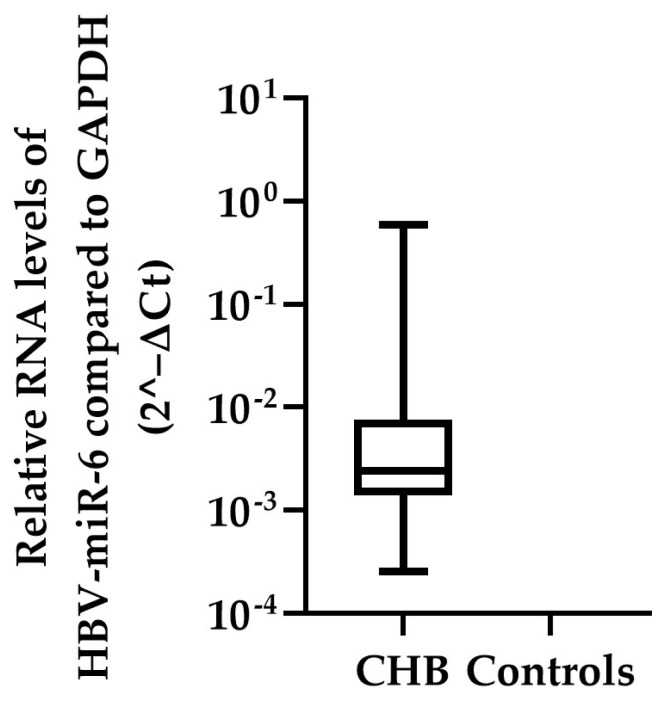
Hepatic HBV-miR-6 expression levels in CHB patients. HBV-miR-6 expression levels were measured in liver tissue derived from 87 CHB patients and 13 HBV negative controls using qPCR. Relative expression levels of HBV-miR-6 were normalized to GAPDH expression using the 2^−ΔCt method; the data are presented as a boxplot and whiskers.

**Figure 4 viruses-14-01223-f004:**
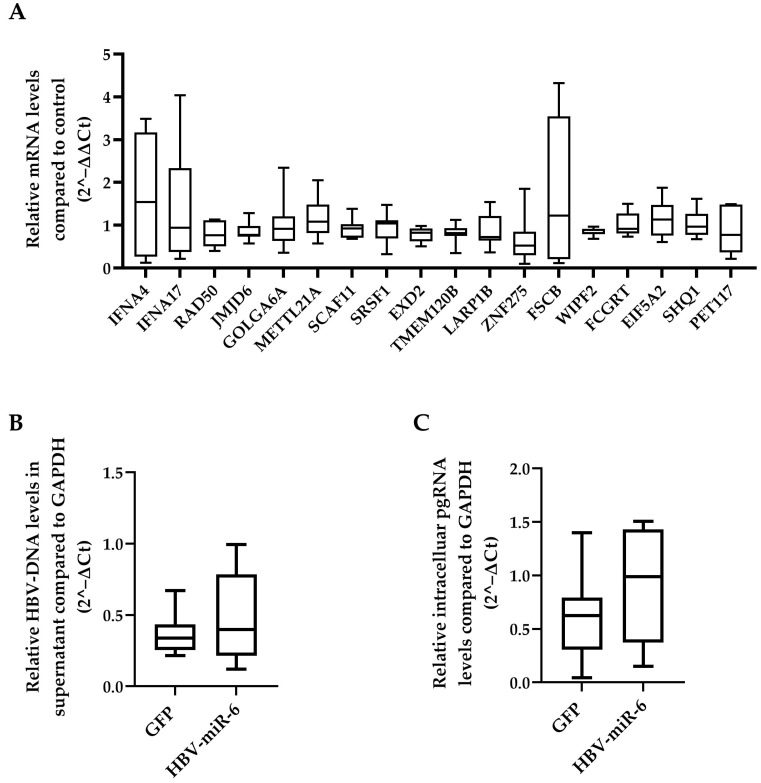
Effect of HBV-miR-6 on the expression of predicted host targets and HBV replication. (**A**) HepG2 cells were transduced with a lentiviral vector expressing HBV-miR-6 or GFP. The expression levels of potential HBV-miR-6 target genes were measured using qPCR at day 6 after transduction. HepG2.2.15 cells were transduced using a lentiviral expressing HBV-miR-6 or GFP. HBV replication was determined at day 6 after transduction using qPCR detecting intracellular pgRNA (**B**) and encapsidated HBV-DNA (**C**) in culture supernatant. Relative gene expression levels were normalized to GFP control using the 2^−ΔΔCt method. Relative expression levels of pgRNA and HBV-DNA were normalized to GAPDH expression using the 2^−ΔCt method and groups were compared using a Mann–Whitney U test. Data are presented as a boxplot and whiskers; graphs represent averages of results from three independent experiments.

**Figure 5 viruses-14-01223-f005:**
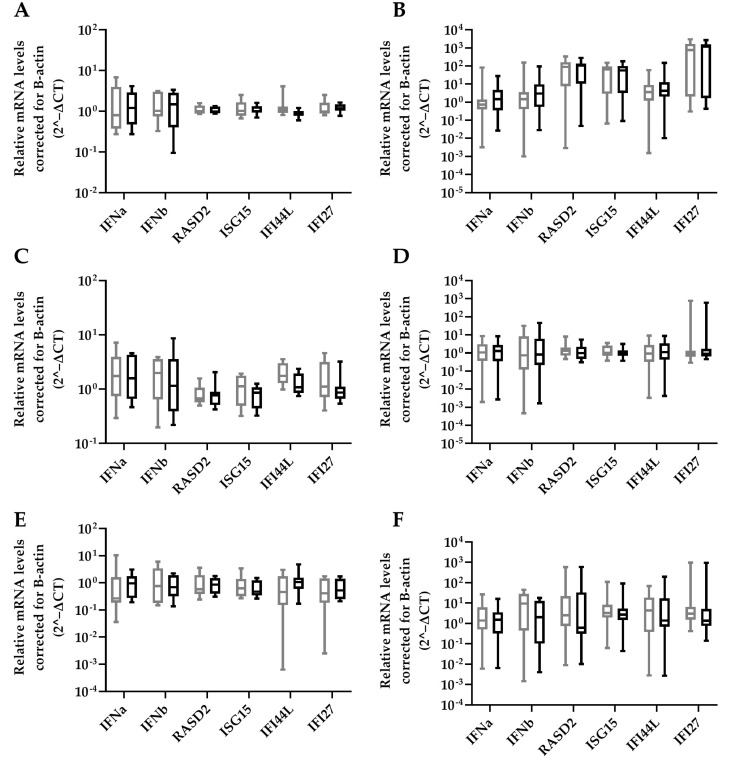
Effect of HBV-miR-6 on innate signaling. HepG2 cells were transduced with a lentiviral vector expressing GFP (grey) or HBV-miR-6 (black). After 48 h, the cells were stimulated with either IFNa (**A**,**B**), R848 (**C**,**D**) or ODN2216 (**E**,**F**). The expression levels of IFNa, IFNb, and ISGs (RASD2, ISG15, IFI44L, IFI27) were measured using qPCR at 6 h (**A**,**C**,**E**) or 16 h (**B**,**D**,**F**) after stimulation. Relative expression levels of mRNA were normalized to B-actin expression using the 2^−ΔCt method and groups were compared using a Mann–Whitney U test. Data are presented as a boxplot and whiskers; graphs represent averages of results from three independent experiments.

**Table 1 viruses-14-01223-t001:** Characteristics of the CHB (n = 97) and control liver (n = 13) cohorts.

Patient Characteristics	CHB Cohort (n = 97)	Control Cohort (n = 13)
Male, n (%)	57 (58.8)	5 (39)
Age, years, mean (SD)	44.5 (11.3)	59 (16)
Region of origin:		
Caucasian, n (%)	29 (29.9)	10 (77)
North African, n (%)	5 (5.2)	
Central African, n (%)	24 (24.7)	
Central Asian, n (%)	3 (3.1)	
Southeast Asian, n (%)	13 (13.4)	
South American, n (%)	17 (17.5)	3 (23)
ALT, U/L, median (IQR)	26 (20–34.5)	30 (24–40)
**Viral characteristics:**		
HBsAg, log10 IU/mL, mean (SD)	3.22 (0.92)	
HBV-DNA, log10 IU/mL, mean (SD)	2.75 (1.15)	
HBV Genotype, n (%):		
A	24 (24.7)	
B	6 (6.2)	
C	4 (4.1)	
D	23 (23.7)	
E	18 (18.6)	
Undeterminable	22 (22.7)	
**Liver Fibroscan and biopsy:**		
Fibroscan value (kPa), mean (SD)	5.4 (2.0)	
Fibroscan IQR range, mean (SD)	0.98 (1.0)	
Ishak fibrosis score, median (IQR)	1 (1–1)	
Modified HAI score, median (IQR)	2 (2–3)	
Steatosis grade, median (IQR)	0 (0–1)	
**Liver Pathology after resection:**		
Colorectal liver metastasis, n		4
Hepatic adenocarcinoma, n		2
Cholangiocarcinoma, n		2
Hepatocellular carcinoma, n		2
Hepatolithiasis, n		2
Hepatic Cystadenoma, n		1

SD, standard deviation; IQR, interquartile range; ALT, alanine transaminase; HBsAg, hepatitis B surface antigen; HAI, Histology Activity Index.

**Table 2 viruses-14-01223-t002:** Correlations between hepatic HBV-miR-6 expression levels and biomarkers of HBV activity and liver damage.

	Rho	*p*-Value
**HBV replication:**		
Plasma HBV-DNA	0.118	0.528
Plasma HBsAg	0.338	0.045
Hepatic HBV-DNA	0.432	0.020
**Liver disease:**		
Modified HAI Grading	0.124	0.528
Ishak fibrosis score	−0.067	0.710
Plasma ALT	−0.147	0.508

Correlations were determined using a Spearman correlation test. Significance was set at *p* < 0.05. ALT, alanine transferase; HAI, hepatic activity index; HBsAg, hepatitis B surface antigen.

## Data Availability

The data presented in this study are available on request from the corresponding author.

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
