# Peer review of "Identification of a Novel HBV Encoded miRNA Using Next Generation Sequencing"

_viruses, 2022, doi:10.3390/v14061223_

Round 1
Reviewer 1 Report
- In section 2.3 authors could not contribute about the PCR conditions.
- Authors could not contributed any immunoblot image of any in vitro expression
- Authors should add conclusion of the study
Author Response
In section 2.3 authors could not contribute about the PCR conditions.
In the current study the NGS of plasma and liver miRNA was performed by the Qiagen miRNA NGS service and therefore description of some in depth methods is missing. We therefore emphasized this by adding an introductory statement in this section.
Authors could not contributed any immunoblot image of any in vitro expression
Due limited patient material availability it is not possible to perform a Notherblot or RIP assay. Instead we have used a specific quantitative PCR to detect the HBV encoded miRNAs in patient material. This PCR uses miRNA specific primers in combination with an adaptor primer. The small size of the final PCR product is confirmed by melting curve analysis. In the cell culture system we have used an antibody based RNA immune precipitation technique to analyze whether the potential HBV derived miRNAs are able to bind to RISC, to confirm that the transcript that we have analyzed is indeed a miRNA. Moreover, controls have been included to confirm that the transcripts we are measuring are indeed derived from HBV.
Authors should add conclusion of the study
We have added a separate conclusion heading in the text and expanded our conclusion.

Reviewer 2 Report
Dear Editor,
The manuscript entitled “Identification of a novel HBV encoded miRNA using Next Generation Sequencing” by Loukachov et al. aim to identify novel HBV encoded miRNAs in plasma and liver tissue samples from chronic hepatitis B (CHB) patients by NGS. The authors identify three potential miRNA like structures transcribed by HBV in liver tissue, and one of them (HBV-miR-6) was recognized by RISC, as determined by RNA immunoprecipitation. The effect of HBV-miR-6 overexpression on HBV replication, the expression of predicted target genes and induction of interferon stimulated genes in two cell lines were also assessed.
Τhe manuscripts’ objects are quite interesting, the manuscript is well-written and could be accepted for publication after major revisions. My detailed comments for the authors to consider are provided below:
- My main concern is that the manuscript title is somewhat misleading. In my opinion the authors do not present the selection of a miRNA. The authors fail to describe adequately the methodology by which they ended up to the candidate miRNAs. Have they used size selection? The selected miRNAs are quite long (longer than 60 nt), therefore are not miRNAs as described in introduction (19-24 nt). Are these their precursor molecules? How are they used and what is the mechanism of action? All these points should be clarified.
- Why the HBV-miR-6 target gene expression was tested only in HepG2 cell line and not to both cell lines?
- All qRT figures would be more informative if presented as box and whiskers instead of bars.
- How many experiments have resulted in the depicted SD? I could not find that information in any figure.
- How is 16 hours time point was chosen for IFN related genes determination? Maybe an earlier time point would be more appropriate? In any case more discussion and references on that subject would benefit the manuscript
- More information concerning the lack of miRNAs detection in plasma and the potential mechanism of HBV-miRNA-6 action should be added to the discussion section.
Author Response
My main concern is that the manuscript title is somewhat misleading. In my opinion the authors do not present the selection of a miRNA. The authors fail to describe adequately the methodology by which they ended up to the candidate miRNAs. Have they used size selection? The selected miRNAs are quite long (longer than 60 nt), therefore are not miRNAs as described in introduction (19-24 nt). Are these their precursor molecules? How are they used and what is the mechanism of action? All these points should be clarified.
The authors fail to describe adequately the methodology by which they ended up to the candidate miRNAs. Have they used size selection?
We agree with the reviewer that the description of the size selection is missing. In the current study the selection of potential miRNAs reads was based on size selection. However as the NGS of plasma and liver miRNA was performed by the Qiagen miRNA NGS service these specifics were not included in the description of the methods. Nevertheless we did add a sentence in the method section mentioning this size selection.
The selected miRNAs are quite long (longer than 60 nt), therefore are not miRNAs as described in introduction (19-24 nt). Are these their precursor molecules? How are they used and what is the mechanism of action?
We apologize for this inconsistency. The hairpin structures showed in figure 1 are indeed not the miRNA but the pre-miRNAs. This hairpin likes structures is between 60–120 nt long and is subsequently cleaved by DICER resulting in a double-stranded RNA between 19 to 25 nucleotides in length. This duplex is subsequently loaded into the RNA-induced silencing complex (RISC), where one of the two RNA strands is degraded. The remaining strand, the mature miRNA, binds to mRNA which can either lead to degradation or repression, or in some cases in upregulation of gene expression.
The hairpin like pre-miRNA structure is referred to as miR follow by a number. The mature miRNAs, which are between 19-24 nt long, are referred as miR follow by a number-3’ or miR follow by a number-5’. We have adjusted the manuscript to eliminate this inconsistency.
Why the HBV-miR-6 target gene expression was tested only in HepG2 cell line and not to both cell lines?
In this study we overexpressed HBV-miR-6 in HepG2 cells and subsequently assessed the expression levels of the predicted targets. Hep2.2.15 cells contain several integrated HBV copies, and therefore these cells produce and excrete HBV particle together with potentially HBV encoded miRNAs including HBV-miR-6. Thus this will influence the expression levels of the potentially targeted genes prior the overexpression of HBV-miRNA-6 in these cell lines.
All qRT figures would be more informative if presented as box and whiskers instead of bars.
We thank the reviewer for this suggestion and have depicted the data as boxplot and whiskers were applicable.
How many experiments have resulted in the depicted SD? I could not find that information in any figure.
We apologize for this missing information and have added the number of independent experiment in the corresponding figure legends.
How is 16 hours time point was chosen for IFN related genes determination? Maybe an earlier time point would be more appropriate? In any case more discussion and references on that subject would benefit the manuscript
We thank the reviewer for this suggestion and have added a 6 hour time point. Similarly to the 16 hour time point, after six hours of TLR agonist stimulation no effect of HBV-miR-6 overexpression on IFN related gene expression was observed. The selection of these time points was based on previous performed studies. This point is added the discussion with corresponding references.
More information concerning the lack of miRNAs detection in plasma and the potential mechanism of HBV-miRNA-6 action should be added to the discussion section.
We thank the reviewer for this recommendation. We tried to measure HBV-miR-6 in plasma but as we failed to measure these expression levels, we did not add this to the manuscript. As this is indeed valuable information, we therefore adjusted the results and discussion section.

Round 2
Reviewer 1 Report
1. In the cell culture system authors used an antibody based RNA immune precipitation technique to analyze whether the potential HBV derived miRNAs are able to bind to RISC. Can authors add its blot image?
Author Response
We have used RNA immune precipitation using a specific antibody against argonaute-2 (AGO2), which is part of RISC to pull down AGO2 and miRNA complex. To determine if the potential HBV miRNAs were associated with RISC, a RT-qPCR specific for the different miRNAs was performed. This method is commonly used to identify protein/RNA interactions and does not involve Nothern blotting. Therefore we are not able to provide a blot image.

Reviewer 2 Report
Dear Editor,
The authors of the manuscript entitled “Identification of a novel HBV encoded miRNA using Next Generation Sequencing” by Loukachov et al. has provided adequate responses to my concerns.
The answer provided in the comment: “The selected miRNAs are quite long (longer than 60 nt), therefore are not miRNAs as described in introduction (19-24 nt). Are these their precursor molecules? How are they used and what is the mechanism of action? All these points should be clarified” should be incorporated in the manuscript text.
Also the autors state that “mature miRNAs, which are between 19-24 nt long, are referred as miR follow by a number-3’ or miR follow by a number-5’. We have adjusted the manuscript to eliminate this inconsistency” but I was unable to find any such miRNA naming in the text. Does that mean that all used miRNAs in experimental section are in fact the precursor molecules?
Overall, the manuscript is well-written and could be accepted for publication after minor revisions.
Author Response
The answer provided in the comment: “The selected miRNAs are quite long (longer than 60 nt), therefore are not miRNAs as described in introduction (19-24 nt). Are these their precursor molecules? How are they used and what is the mechanism of action? All these points should be clarified” should be incorporated in the manuscript text.
We adjusted the text in the manuscript to state that these are indeed pre-miRNAs. Furthermore, we incorporated the above mentioned points in the introduction section.
Also the autors state that “mature miRNAs, which are between 19-24 nt long, are referred as miR follow by a number-3’ or miR follow by a number-5’. We have adjusted the manuscript to eliminate this inconsistency” but I was unable to find any such miRNA naming in the text. Does that mean that all used miRNAs in experimental section are in fact the precursor molecules?
We would like to clarify the last question asked by the reviewer. In this study we wanted to mimic miRNA overexpression as naturally as possible and therefore chose to overexpress the HBV-miR-6 sequences. To this extent the miR-6 sequences flanked by the 100 base pair up- and downstream HBV sequence was cloned into a lentiviral vector. The flanking regions of the miRNA were included to ensure the correct secondary structure folding and processing by RISC. HepG2 cells are efficiently transduced by lentiviral vectors and this ensures long term expression of the miRNA. The reviewer is correct that in this study we overexpressed the miRNA hairpin structure to ensure the best possible biological mimic. Another possibility is to overexpressed a synthetic siRNA containing mature miRNA duplex sequences. However this is highly artificial and the overexpression efficiently is low as these siRNAs should be transferred into the cell by transfection which is highly inefficient in HepG2 cells. Moreover, the overexpression by transfection is short live as these siRNAs are lost during cell proliferation and siRNAs are degraded.
